# High-Quality 3D Visualization System for Light-Field Microscopy with Fine-Scale Shape Measurement through Accurate 3D Surface Data

**DOI:** 10.3390/s23042173

**Published:** 2023-02-15

**Authors:** Ki Hoon Kwon, Munkh-Uchral Erdenebat, Nam Kim, Anar Khuderchuluun, Shariar Md Imtiaz, Min Young Kim, Ki-Chul Kwon

**Affiliations:** 1School of Electronic and Electrical Engineering, Kyungpook National University, Daegu 41566, Republic of Korea; 2School of Information and Communication Engineering, Chungbuk National University, Cheongju 28644, Republic of Korea

**Keywords:** light field, light-field microscopy, depth estimation, fine-scale shape measurement, 3D visualization, integral imaging

## Abstract

We propose a light-field microscopy display system that provides improved image quality and realistic three-dimensional (3D) measurement information. Our approach acquires both high-resolution two-dimensional (2D) and light-field images of the specimen sequentially. We put forward a matting Laplacian-based depth estimation algorithm to obtain nearly realistic 3D surface data, allowing the calculation of depth data, which is relatively close to the actual surface, and measurement information from the light-field images of specimens. High-reliability area data of the focus measure map and spatial affinity information of the matting Laplacian are used to estimate nearly realistic depths. This process represents a reference value for the light-field microscopy depth range that was not previously available. A 3D model is regenerated by combining the depth data and the high-resolution 2D image. The element image array is rendered through a simplified direction-reversal calculation method, which depends on user interaction from the 3D model and is displayed on the 3D display device. We confirm that the proposed system increases the accuracy of depth estimation and measurement and improves the quality of visualization and 3D display images.

## 1. Introduction

Integral photography (IP) technology was first reported by Gabriel Lippmann [1] in the early 20th century. By the 1990s, the technology had expanded to capture simultaneous multi-view information about a three-dimensional (3D) scene through a lens array. Many notable research results have been published [2]. The process of describing the luminosity of captured light in an IP system in mathematical terms has been named plenoptic [3] or light-field (LF) [4] technology.

LF imaging techniques, especially based on integral photography, can acquire realistic 3D information of a 3D scene concurrently through a lens array. This can be reconstructed into individual viewpoints or refocused depth-slices through post-processing. An LF camera uses an image-pickup device, in which a microlens array (MLA) is added between the main lens and the sensor. Although many perspective images can be acquired through MLA, the parallax is very small and the spatial resolution of each perspective image decreases in proportion to the number of elemental lenses in the array [5]. High numerical aperture objectives can capture rays with broad angular data. Using this advantage, Levey et al. reported a light-field microscopy (LFM) structure in which MLA is inserted into the image plane of a conventional optical microscope [6].

Unlike other 3D optical microscopy technologies, such as confocal and holographic microscopy, LFM can obtain 3D information (parallax, depth information, etc.,) about the sample through the MLA and the non-coherent illumination. It can be reconstructed with accurate color 3D visualization [7]. In addition, spatial and angular information about the specimen can be collected in real time. Note that, the acquired 3D information of the specimen which is encoded by LFM is much more than conventional optical microscopes. Despite these advantages, the depth of field and viewing angle are insufficient for comfortable viewing, and the lighting of the environment greatly affects the acquired images; therefore, it is difficult to reconstruct fine-quality 3D visualizations [8]. Several adaptations have been proposed to enhance the limitations of LFM, including improvements in the optical system, resolution, depth of field (DOF), and 3D visualization [9,10,11,12]. However, due to the fundamental limitation of the MLA f-number and the poor illumination conditions of the microscope, it has not yet been possible to produce a satisfactory 3D reconstruction image. Indeed, only poor two-dimensional (2D) LFM images, encompassing the field of view in the orthographic-view image and DOF in-depth slice image of the sample, have been published [13]. Although various approaches have shown potential for DOF improvement [14,15], the only way to improve the resolution is to increase the number of elemental images [16], a quantity that is proportional to the resolution, when the LFM unit with a focused structure is utilized.

Since the LFM image contains 3D information on the captured sample, it is possible to obtain a depth map with 3D surface information of the sample by analyzing the correspondence and focal information [17,18,19]. Additionally, many deep-learning-based LF depth estimation methods have been studied and have shown good depth estimation performance [20,21,22]. Recently, Kwon et al. used an LFM system to acquire both LF and 2D images. The depth information was obtained from the LF and combined with 2D images to create 3D model data, which was then displayed on a high-resolution LF display [23,24]. However, there is a limit to providing even measurement information due to problems in the accuracy of the depth estimation data.

In this paper, we propose and demonstrate an LF microscopy 3D display system that improves the resolution of reproduced images and provides nearly realistic measurement information. A matting Laplacian-based LFM depth estimation algorithm by focus analysis was used for realistic surface estimation and object measurement in the LFM images. In the proposed system, although poor LF images of the samples can be acquired through use of LFM, quite realistic 3D surface data (not the relative depth value) can be regenerated by matching the pre-calculated measurement information with the estimated depth information of the specimen. To visualize the sample, a 3D model was generated using the depth information and a high-resolution 2D image. In the 3D model, a new element image array was produced through a simplified direction-reversal calculation (DRC) method and displayed directly on a display device.

## 2. Principle of Light-Field Microscope

IP technology was originally a 3D display technology that uses a lens array but has recently been applied as an efficient way to acquire LF images. In conventional optical microscopy, at any specific location on the sample, all rays are imaged by the camera pixel, and directional information is lost. Inserting the MLA into the microscopic unit, as shown in Figure 1, allows all rays from a given location in the sample to be stored as an elemental image array (EIA). Note that the LFM units of the proposed system are based on the focused MLA structure with an infinity-corrected optical system, where the optical structure of LFM differs depending on where the MLA and intermediate-view image plane are located [5,6].

For a point *P* on the sample, all rays traveling in different directions can be expressed in the 5D plenoptic function consisting of three spatial coordinates (x,y,z) and two angular parameters (θ,φ). It can be expressed as L(s,t,u,v), which is a 4D LF function corresponding to the sensor plane through the image plane where the intermediate image plane is located between the objective lens and the MLA that are separated by the corresponding focal distances. The LF microscope captures images with the 4D LF function, that is, the EIA. By analyzing this, it is possible to reconstruct various viewpoint images and variable focus images of the specimen.

Although LFM acquires 3D information from micro-samples, the process of reconstructing the acquired information back to a 3D image is very complicated. To reconstruct a high-quality 3D image, first, the acquired image must be of good quality, and the 3D information must be reliably encoded. However, due to the use of a lens array, it is almost impossible to capture high-quality LF images in the LFM system. Moreover, the 3D information, especially the depth information, is not certain as EIA is obtained, through the MLA, from micro-objects. Previously, when the 3D model was regenerated through the deep-learning-based LF depth estimation model, only the approximated shape could be estimated, meaning that the 3D depth information was still unclear due to the constraints of the LFM image and the domain gap problem [23].

## 3. Materials and Methods

### 3.1. Optical Design

The hardware units of the proposed system, including the LFM system and 3D display, are shown in Figure 2. The LFM system consists of two different optical microscopes: a stereo microscope Olympus SZX7, and a biological microscope BX41; a camera and an MLA are installed in each of them. Both LFMs can acquire the EIA and 2D images depending on whether the MLA is placed between the objective lens and the camera. The reason for the use of different LFMs is because of the magnification: the SZX7-based LFM acquires the 3D information of a specimen through low magnification, and the BX41-based LFM is used for high magnification, depending on the conditions. The specifications for all devices are listed in Table 1. Using this LFM system, LF images were acquired from four micro-samples: a grass seed, a micro gear, a balsam seed, and a fruit fly. Figure 2a shows the LFM unit and Figure 2b,c show the high-resolution 2D images and EIAs for each specimen obtained by the LFM system. The EIAs include 76 × 76 elemental images after the regions of interest are selected by removing the outer black portions; the resolution of the entire image is 4000 × 4000 pixels. This is identical to the resolution of the 2D images. Note that, even though the MLA consists of 100 × 100 lenses, the camera lens is matched to capture the middle part of the EIA, which is the active area of information of the specimen to be gathered. This active area was composed of 76 × 76 elemental images during the experiment; the horizontal size was cropped to be the same as the vertical size.

The 3D display unit consists of a display device and a lens array. The display unit is shown in Figure 2d. A lens array is attached to the 6-mm-thick acrylic plate and mounted in front of the display device, according to the integral imaging display technique. When the EIA is generated, following the DRC-based approach, it is exhibited on the display device, and a natural-view full-color and full-parallax 3D visualization of the corresponding sample is successfully reconstructed.

### 3.2. Depth Estimation Method and Measurement

Although the LFM image contains the entire 3D information of the specimen and can be reconstructed as refocused depth-slices computationally, as shown in Figure 3, it is difficult to estimate the exact shape and depth of the sample, because it includes resolution degradation and a lot of noise due to using a lens array and the poor lighting environment. To estimate realistic depths of the sample in the LFM, we propose a depth estimation method, based on reliable area information in the initial depth map generated through the focus measure analysis of the LFM image and spatial affinity information obtained through the matting Laplacian. The relative depth map is converted into a real-distance depth map using the fitting coefficient obtained through the pre-performed distance calibration process. In total, this LFM depth estimation method comprises digital refocusing, depth-map estimation, and depth-map conversion processes.

First, the LFM image is converted into a refocused image set through a digital refocusing process based on the 4D Fourier slice theorem [25]. As shown in Equation (Equation 1), the EIA can be converted into refocused images through 2D slices and inverse transformation in the Fourier domain of the 4D LF. Images focused on different depths are generated through the inverse transformation of 2D slices with different trajectories.
(1)$$P_{\alpha }\equiv F^{-2}\circ P_{\alpha }\circ F^{4}$$
where $F^{4}$ is the 4D Fourier transform, $P_{\alpha }$ is the slicing operator and $F^{-2}$ is the inverse 2D Fourier transform. In the digital refocusing process, the refocused image set used for depth estimation is obtained from another virtual image plane according to the alpha coefficient (α), defined by the distance ratio relationship between each image plane and the MLA plane. The refocused image set is an image set reconstructed according to the linear focal length change information corresponding to α, as shown in Figure 3.

Figure 4 shows the LFM depth estimation method. First, in the focus measuring step, the initial depth map is extracted by analyzing the focal area of each image through the Laplacian-operator-based focus measurement process in a refocused image set [26]. In this step, focus measures are performed on all images in the image set; an initial depth map is generated by indexing the corresponding α value of the maximum focus measure value for each pixel (area). In addition, the central viewpoint image, which is the same viewpoint as the refocused image set, is reconstructed from the EIA.

Next, a sparse depth map and data precision are obtained in which only areas with high confidence are filtered from the initial depth map through the reliability analysis process, based on the focus measure value. The central view image of the LF image is converted into a matting Laplacian matrix via the matting Laplacian process [27]. This matrix reflects the spatial similarity between pixels in the input image; the size of the matrix is also determined by the size of the input image. The depth map reconstruction process is performed based on this information [28,29]. To convert the sparse depth map to the reconstructed depth map with global spatial coherence restored, maximum a posteriori (MAP) estimation is utilized based on the matrix. The MAP estimation can be solved by minimizing the cost function after prior pre-modeling on spatial coherence between the depth image and the color image, as shown in Equations (Equation 2) and (Equation 3). To acquire the reconstructed depth image, the global minimum of Equation (Equation 3) can be obtained by solving a system of linear equations, such as Equation (Equation 4). The reconstructed depth map represents relative depth information (based on focal length).
(2)$$D^{*}=\underset{D}{\mathrm{argmax}}\;\left(p\left(D|I^{Set}\right)\right)$$
(3)$$-log\left(p\left(I^{Set}|D\right)p\left(D\right)\right)={\left(\tilde{d}-d\right)}^{T}\Lambda \left(\tilde{d}-d\right)+d^{T}Ld$$
(4)$$\left(L+\Lambda \right)d=\Lambda \tilde{d}$$
where *D* is the depth image, $I^{Set}$ is the corresponding color image set, $\tilde{d}$ is the sparse depth map, Λ is the data precision, *L* is the matting Laplacian matrix and *d* is the reconstructed depth map (optimal depth image).

The pixel intensity of the reconstructed depth map corresponds to the α value. The α variation corresponds to the displacement of the image space, which has a linear relationship with the displacement of the object space [18]. Therefore, if the relational expression between the measured value and the estimated intensity value of the depth map is obtained, the depth map in α value units can be converted into the depth map in measured value units. As shown in Figure 5, a step-shaped 3D board is used to perform the pre-distance calibration. Using the proposed method, the depth map of the 3D board is estimated and the ROI is set for each floor of the stairs in the depth map. Representative values close to the average of each ROI are extracted and linear curve fitting is performed. The fitting coefficient obtained through pre-distance calibration is used to convert the depth map of specimens into a real-distance depth map, as shown in Figure 5.

### 3.3. 3D Model Generation and 3D Visualization

Figure 6 shows the entire process of the EIA generation for 3D visualization. Firstly, the point cloud object is obtained from the estimated depth data and high-resolution 2D image (including color and texture), which meets the requirements of comfortable viewing that is as clear as a 2D image. The depth image corresponding to the 2D image is resized for matching the resolution of the 2D image. In order to obtained a smoother point cloud model, the resized depth map is interpolated by utilizing the surface-interpolating corner vertices and boundary curves to fill the holes. A high-quality 3D model is generated by obtaining the RGB-D point cloud object from the depth map, including the 3D surface and the 2D image, including color and texture. After that, the EIA is generated using the 3D object with high-quality, as shown in Figure 6c, by applying the DRC method [30]. This is based on the backward rendering of computer-generated integral imaging, where depth and color information of the 3D model is obtained according to the layered depth data and the main parameters of a virtual lens array and display device are given by the user.

In the generation of EIA, the light is propagated from the EIA plane to the 3D model through the corresponding single elemental lens where it intersects the object points. Consequently, the color information of an arbitrary object point intersecting the propagated light is stored at the corresponding pixel of the EIA. Note that the total resolution of the EIA plane is equal to the number of light propagations; it does not require a checking process for the ray intersection of all object points with every elemental lens, unlike conventional integral imaging techniques. If there is a 3D object point intersecting the light ray, the color information of the object point is stored in the EIA plane and the computation for such a ray will not be implemented in the next depth layer. Therefore, the DRC-based method reduces the calculation time and considers the occlusion effect while providing a feature of an independent elemental image that leads to sufficient computation for parallel computing. The reduction in the EIA computation time is beneficial to the entire system even though the EIA generation cannot support real-time rendering. Moreover, users can carry out functions, such as rotate and/or zoom in/out, and an updated EIA is generated via DRC-based rendering. More viewpoints are demonstrated compared with conventional methods, including nearly realistic depth information of the 3D scene via the basic integral imaging display.

## 4. Result and Discussion

### 4.1. Depth Estimation

Figure 7 shows the comparative results by applying different LF depth estimation methods to four specimens when studied with LFM. Figure 7a shows high-resolution 2D images and Figure 7b shows the central-view images of the LF images. Figure 7c presents the depth estimation method results, based on the cost volume generated by analyzing the correspondence of the LF image [17]. The approximate shape can be estimated, but the detailed internal estimation is inaccurate. Figure 7d displays the result of a fully convolutional neural network using epipolar geometry for depth from light-field images (EPINET), which estimates depth with a pre-trained deep learning model [20]. Again, only the approximate shape can be estimated—there is a blur at the boundary and there is a lot of noise. Figure 7e shows the result of refining the initial depth map, estimated through focus measure, to the depth map using the graph cut algorithm [31] as a focus analysis-based depth estimation method. It can be seen that depth estimation is possible to some extent in the focused edge area, but depth estimation of the inner area of the object is not properly performed. The proposed LFM image has a very low resolution of 76 × 76 per viewpoint and the image is severely degraded by the microscope illumination and MLA. Therefore, it is difficult to analyze correspondence or focus in the LFM image, and, due to the difference in the domain gap from the general LF image, the performance limit is clear when the existing LF depth estimation method is applied as it is. However, as shown in Figure 7f, the depth estimation method used in this study shows that, even in poor LFM images, the depth can be estimated in some detail, with clear boundaries between the background and the object, as well as a clear object shape. These experimental results clearly show that, when estimating depth in LF images under poor conditions, restoring a highly reliable sparse depth map to a dense depth map through the analyzed image information is a suitable strategy. Additionally, the total processing time of the depth estimation, including the initial depth map and reconstructed depth map estimation, was approximately 3.36 s.

The proposed method for measuring LF depth maps was compared with current approaches. A discrete entropy [21] test was used to quantify it, alongside the correspondence analysis-based method, EPINET, and the focus analysis-based method. The more obvious the difference between the background and foreground of the estimated depth map is, the higher the contrast will be and the higher the discard entropy value. Figure 8 illustrates the quantitative results of discrete entropy and that the proposed depth map demonstrated better contrast than existing methods. According to the proposed method, the discrete entropy values for the grass, micro gear, garden balsam seed, and fruit fly were 7.04, 6.68, 7.19, and 8.01, respectively. The corresponding values were 3.24, 3.72, 4.79, and 4.3 for the correspondence analysis-based method, 5.14, 6.11, 4.74, and 5.83 for the EPINET, and 3.36, 4.4, 5.78, and 3.92 for the focus analysis-based method, respectively.

### 4.2. Measurement Analysis

For distance calibration, a 3D board in the form of stairs, with a floor height of 1 mm, was designed as shown in Figure 9a,b. It was printed out with a 3D printer and used as a fitting board. Linear curve fitting was performed between the measured value and the α value using the 3D board. The fitting coefficient was 6.492. Figure 9d shows the output of the 3D model through the depth map being converted to the real-distance value using the fitting coefficient. Using the sliced graph of the 3D model, it can be confirmed that the estimated 3D model approximates the actual value. A triangular column of 1 mm height, a cylinder of 2 mm height, and a square column of 3 mm height were printed on a 3D printer and used as test boards for quantitative analysis of measurement accuracy, as shown in Figure 9e,f. As depicted in Figure 9h, the test boards were converted into 3D models using the same fitting coefficients and the ROI was specified for each test board to analyze the measurement errors. The triangular column had a mean height of 1.018 mm with a mean error of 0.032 mm, the cylinder had a mean height of 1.925 mm with a mean error of 0.190 mm, and the square column had a mean height of 2.886 mm with a mean error of 0.122 mm. The approximate error range was 0.1–0.2 mm.

Figure 10 shows the result of converting the micro gear height data into a real-distance 3D model. The height of the gear is about 3 mm, and the height of the 3D model of the proposed method is estimated as a similar value. Considering that the specification of the 3D printer has an output error of 0.1 mm, it can be confirmed that measurement and trend analyses are possible to some extent, although it is not precise even with the simple proposed method.

### 4.3. 3D Visualization Based on the 3D Model

Based on the real-distance depth images and initially captured high-resolution 2D images, 3D point cloud models were regenerated. The object points of each model were 558,421 for the grass seed (87 layers), 553,383 for the micro gear model (70 layers), 577,596 for the balsam seed (101 layers), and 561,170 for the fruit fly (101 layers). The upper row of Figure 11 shows the appearance of the 3D point cloud models.

The new EIAs were generated within 2160 × 2160 pixels, according to the specifications of the lens array and display device, while the lens array was mounted in front of the display device. Figure 11, Figure 12 and Figure 13 show the generated EIAs, the various viewpoints of the 3D visualizations, and the zoomed-in/zoomed-out images for four samples, except for the appearance of the 3D point cloud models. The EIAs were generated from the DRC-based approach by applying graphic processing unit parallel computing. The total processing time for EIA generation was 1.01 s for the grass seed, 0.9 s for the micro gear, 1.05 s for the balsam seed, and 1.04 s for the fruit fly. Note that, the experimental environment was a Windows 10 64-bit operating system, with an Intel Core i7-8700 central processing unit at 3.2 GHz, 16 GB RAM, and NVIDIA GeForce GTX 1080Ti. The viewing angle of the 3D display was approximately $13^{{}^{\circ}}$. From the reconstructed 3D images, it can be verified that the quality was close to the 2D images, and, importantly, that the obtained 3D information was decoded reliably. In addition, the user could see the various viewpoints of the specimens as well as the zoomed-in/zoomed-out images. More detailed results can be confirmed in the Appendix A.

Finally, to verify the improved quality of the 3D visualizations, we evaluated the central viewpoints of the 3D images ($0^{{}^{\circ}}$ viewpoint of Figure 12) using the natural image quality evaluator (NIQE) as the non-reference image-quality assessment method, and the structural similarity index measure (SSIM) as the reference image-quality assessment method [32,33]. The 2D images (Figure 7a) were utilized as the reference for SSIM. Figure 14 shows the 3D images used for image-quality evaluation. Visually, it can be confirmed that the proposed 3D images have a clearer and higher quality than the 3D image from the original EIA.

The SSIM and NIQE values in Figure 15 show that the proposed 3D display method provided higher quality than the conventional approach. Note that, the higher SSIM value (closer to 1) and the lower NIQE value indicate improved image quality. From the graphs of Figure 15b, it can be seen that the proposed 3D images were scored similarly to the 2D images, confirming that the reconstructed 3D images had a quality close to the high-resolution 2D images. Note that, this is an approximate evaluation rather than a precise evaluation, but the results presented indicate that the quality of the reconstructed 3D images correlates with the color and texture of the corresponding 2D images.

## 5. Conclusions

In this paper, we proposed and demonstrated an LFM 3D-display system that improves the resolution of reproduced images and provides realistic sample measurement information. Both high-resolution 2D images and 4D LF images can be acquired with an existing optical microscope device, while user interaction is applied in visualization by reconstructing a high-quality 3D model. Finally, the model was displayed through a 3D-display device. A matting Laplacian-based depth estimation method was proposed to extract depth information. 3D information from the LF images and measurement information in the depth direction was calculated. The proposed depth estimation method extracted nearly realistic depth information despite the poor input images. In addition, the measurement information was found to be more nearly realistic when expressed as an absolute value with a reference rather than as relative depth. The depth information was converted into a PC, and the colors and textures of the high-resolution 2D images were converted into a high-definition 3D model. There was no limitation on the resolution and image quality, and user interaction, such as rotation and zoom in/out, was possible within a certain angle of view. Finally, based on the user interaction, the EIA was regenerated through the DRC method and displayed on the 3D integral imaging display device. The proposed system showed a good enough improvement that the 3D visualization and display were comparable to the 2D high-resolution original images. Further research will focus on improving the computation speed of the overall procedure.

## Figures and Tables

**Figure 1 sensors-23-02173-f001:**
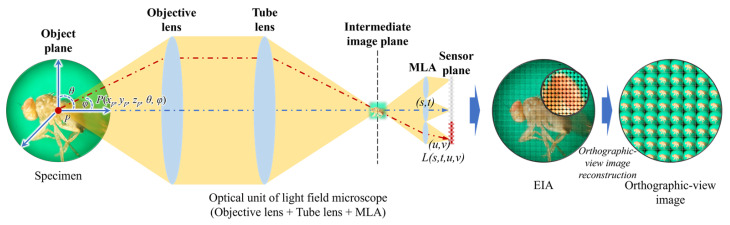
The main acquisition and reconstruction processes of the LFM system.

**Figure 2 sensors-23-02173-f002:**
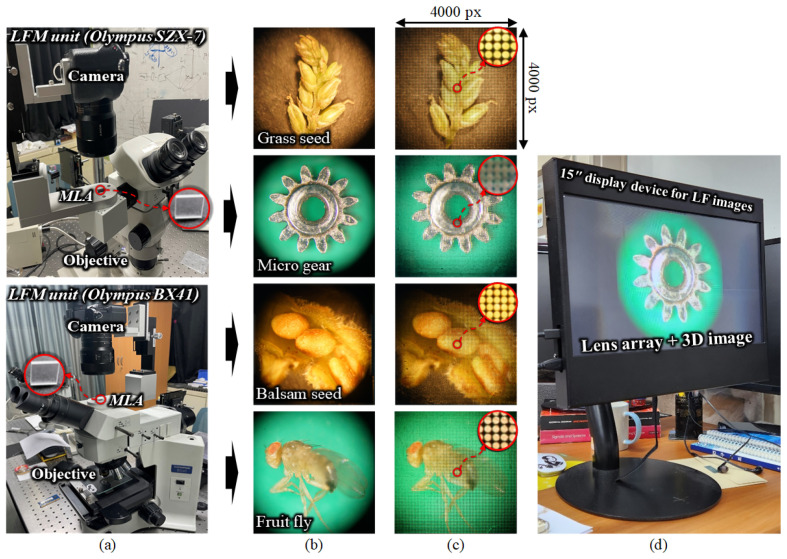
(**a**) Stereo LFM unit for low magnification (top) and biological LFM unit for high magnification (bottom), (**b**) high-resolution 2D images, (**c**) corresponding EIAs of the grass seed, micro gear, balsam seed, and fruit fly samples, and (**d**) a prototype of the 3D LF display unit.

**Figure 3 sensors-23-02173-f003:**
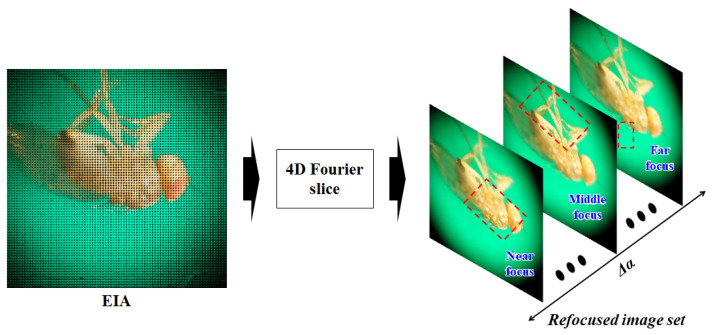
Digital refocusing process.

**Figure 4 sensors-23-02173-f004:**
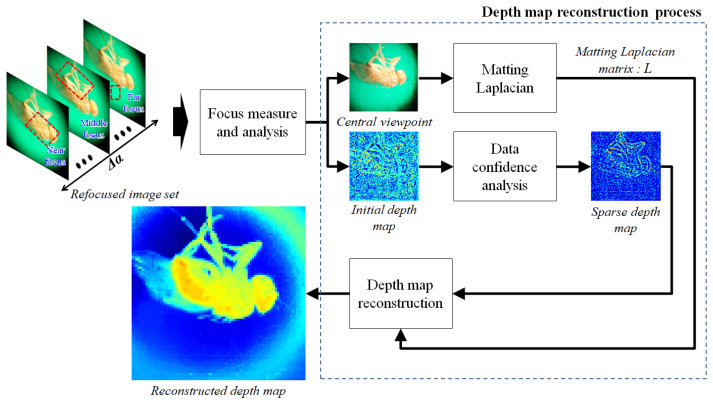
Depth estimation method.

**Figure 5 sensors-23-02173-f005:**
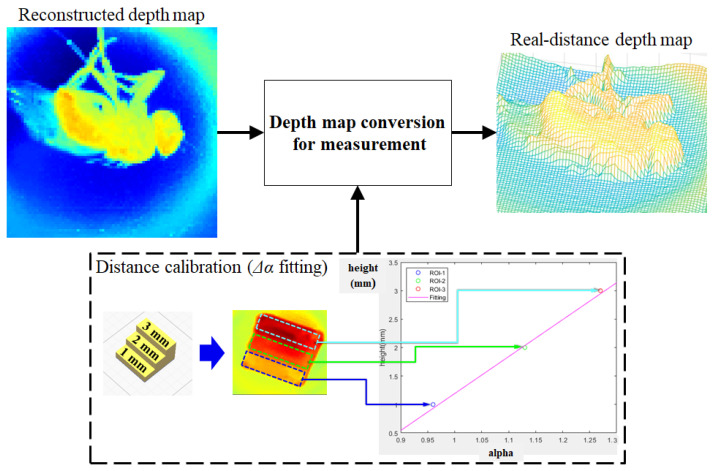
Depth map conversion process via distance calibration.

**Figure 6 sensors-23-02173-f006:**
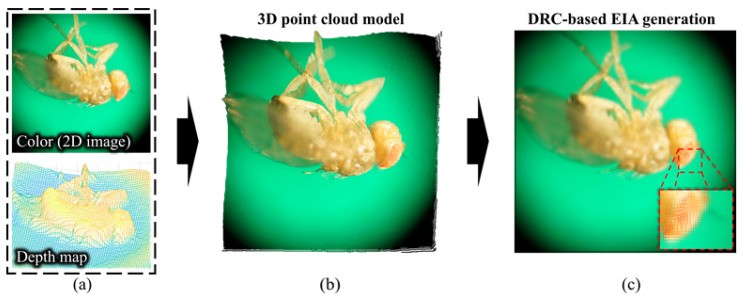
Process of EIA generation for 3D visualization; (**a**) depth and color information, (**b**) generated 3D point cloud, and (**c**) corresponding EIA based on DRC.

**Figure 7 sensors-23-02173-f007:**
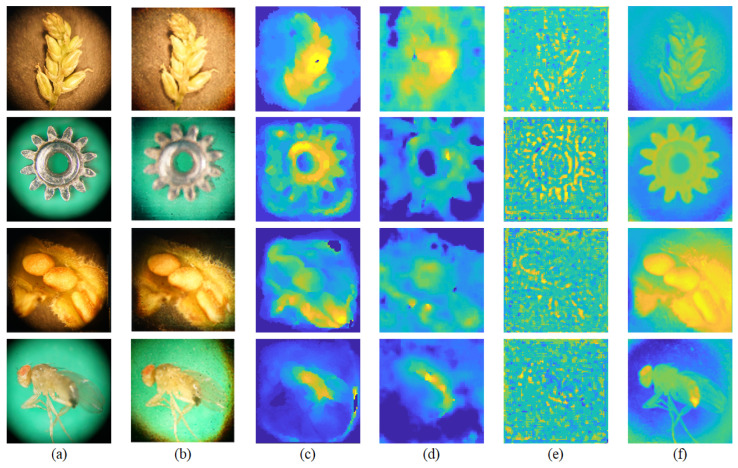
Results of estimated depth information for LF images; (**a**) high-resolution 2D images, (**b**) central-view images, (**c**) correspondence analysis-based method, (**d**) EPINET, (**e**) focus analysis-based method, and (**f**) proposed method.

**Figure 8 sensors-23-02173-f008:**
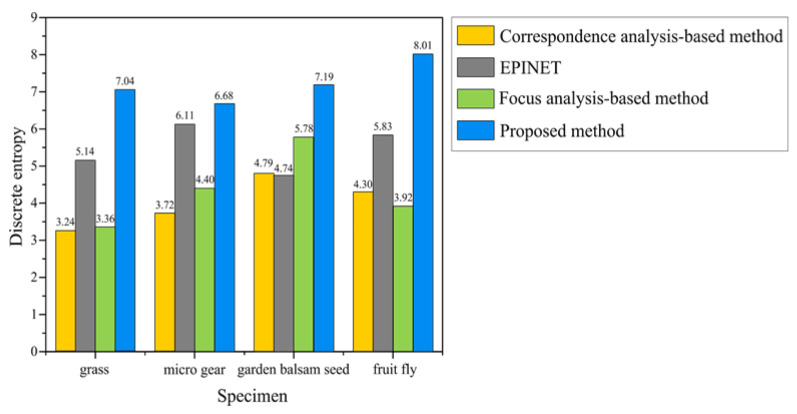
Comparison of the quantitive results of discrete entropy values.

**Figure 9 sensors-23-02173-f009:**
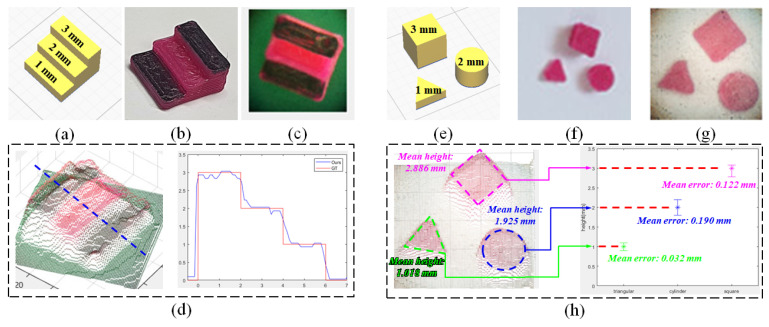
Measurement analysis; (**a**) 3D modeling for distance calibration, (**b**) printed 3D board, (**c**) central-view image of a 3D board, (**d**) real-distance 3D model and slice graph, (**e**) 3D modeling for measurement analysis, (**f**) printed test boards, (**g**) central-view image of test boards, and (**h**) real-distance 3D model and measurement analysis graph.

**Figure 10 sensors-23-02173-f010:**
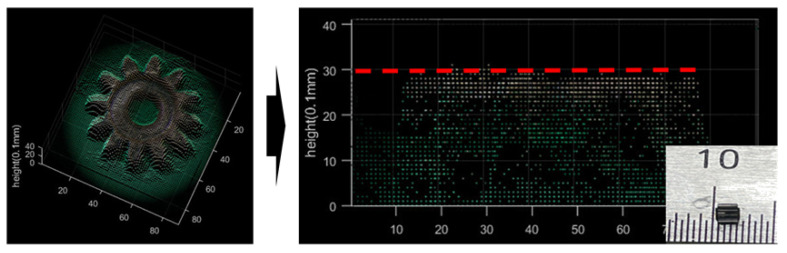
Real-distance 3D model of micro gear.

**Figure 11 sensors-23-02173-f011:**
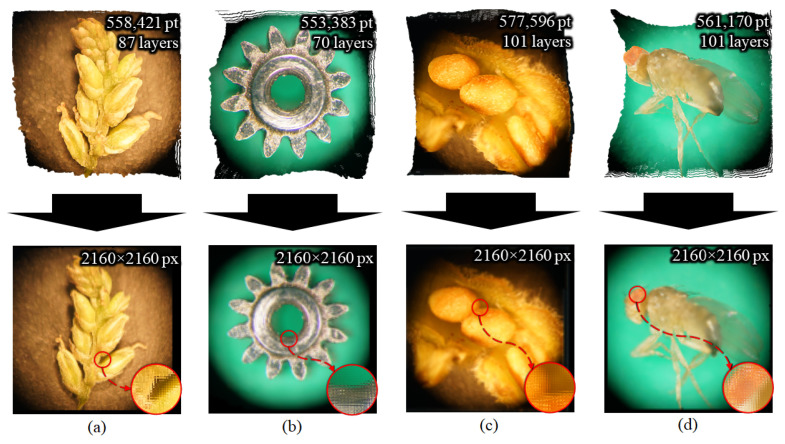
From the 3D point cloud models (top row), the new EIAs were generated through the DRC-based approach (second row) for (**a**) a grass seed, (**b**) a micro gear, (**c**) a balsam seed, and (**d**) a fruit fly.

**Figure 12 sensors-23-02173-f012:**
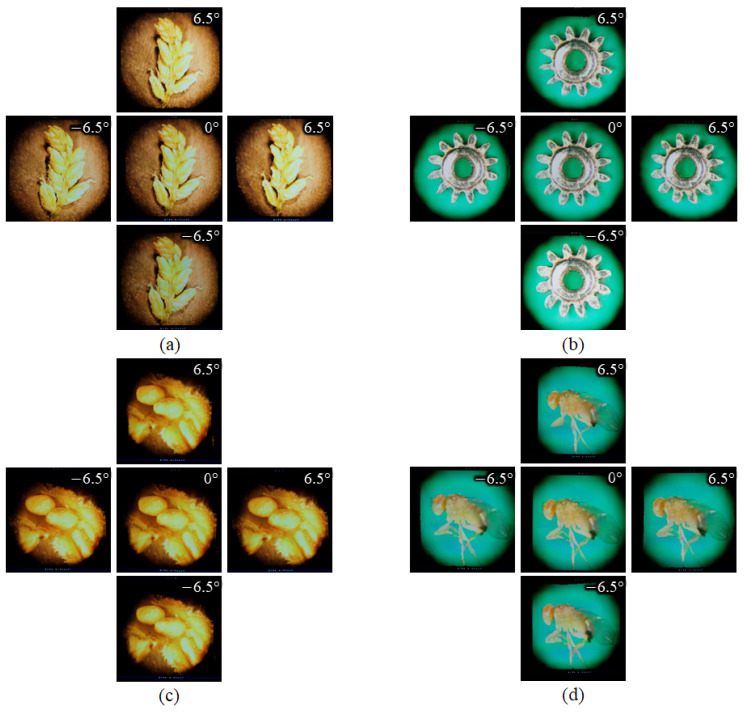
Five viewpoints of the reconstructed 3D images from the newly generated EIAs, for (**a**) a grass seed, (**b**) a micro gear, (**c**) a balsam seed, and (**d**) a fruit fly.

**Figure 13 sensors-23-02173-f013:**
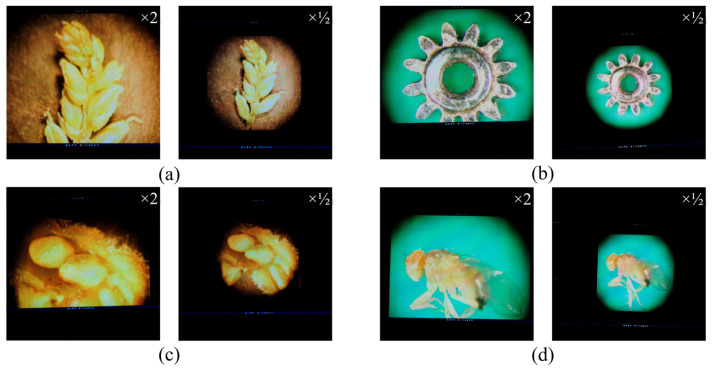
The zoomed-in/zoomed-out visualizations (scale factor = 2) for (**a**) a grass seed, (**b**) a micro gear, (**c**) a balsam seed, and (**d**) a fruit fly.

**Figure 14 sensors-23-02173-f014:**
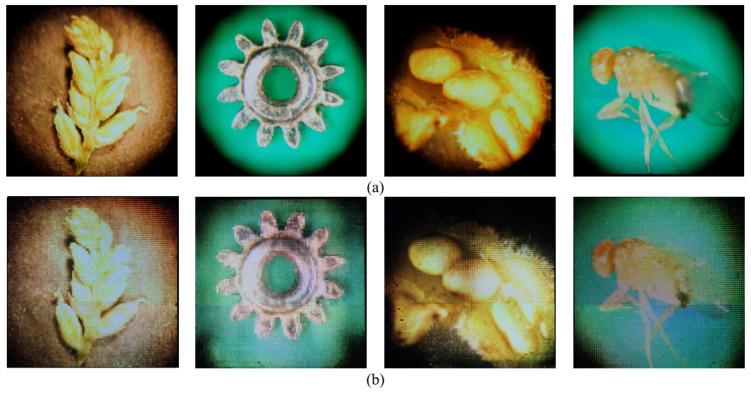
Result of 3D-image capture; (**a**) 3D image using proposed EIA, (**b**) 3D image using original EIA.

**Figure 15 sensors-23-02173-f015:**
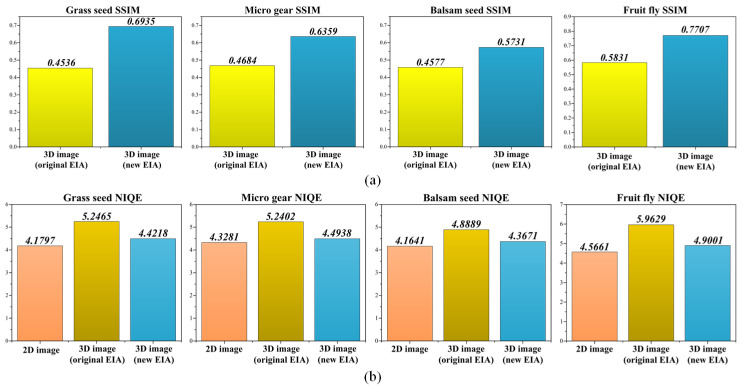
Quantitative evaluation result for comparison; (**a**) SSIM graph, (**b**) NIQE graph.

**Table 1 sensors-23-02173-t001:** Device specifications for the implemented system.

Devices	Indices	Specifications
Microscope Device	For low magnification (SZX-7)	Objective ×1, NA = 0.1
		(Working distance 90 mm)
		Zoom body 7:1
		(Magnification range of 8×–56×)
	For high magnification (BX41)	Objective ×10, NA = 0.3
		(Working distance 10 mm)
MLA	Number of lenses	100 × 100 lenses
	Elemental lens pitch	125 µm
	Focal length	2.4 mm
Camera	Model	Sony α6000
	Sensor resolution	6000 × 4000 pixels
	Pixel pitch	3.88 µm
PC	CPU	Intel i7-8700 3.2 GHz
	Memory	16 GB
	Operating system	Windows 10 Pro (64-bit)
Display device	Screen size	15-inch (345 × 194 mm)
	Resolution	4K (3840 × 2160 px)
	Pixel pitch	0.089 mm
Lens array for 3D display	Focal length	3.3 mm
	Elemental lens pitch	1 mm
	Number of lenses of lens array	345 × 194 lenses
	Thickness of acrylic plate	6 mm

## Data Availability

Not applicable.

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
