# Peer review of "High-Quality 3D Visualization System for Light-Field Microscopy with Fine-Scale Shape Measurement through Accurate 3D Surface Data"

_sensors, 2023, doi:10.3390/s23042173_

Round 1

Reviewer 1 Report

In this manuscript, the authors propose a method for the generation of images to project in light field display, from the capture of microscopic samples through a light field microscope. The method is based on the estimation of a depth map of the samples and its combination with the 2D image, to generate a 3D point cloud. From this point cloud, the elemental images array for the light field (LF) display is computed through a back-propagation algorithm.

This way of proceeding to generate the image to project onto the LF display has already been proposed. The novelty might lie in how the depth map of the sample is estimated, which, although very simple, seems to provide satisfactory results.

The paper is written in a very confusing way and there is a great number of errors and unclear technical points in the explanation, above all in the sections about the LF microscope. All the issues are reported below.

1.      In the abstract, the authors say “Our approach can simultaneously acquire high-resolution two-dimensional (2D) and light field images”.

But, in Section 3.1, the authors say that the system can acquire the light field images or the 2D images whether the microlens array is inserted in the optical path or not. So, is it really simultaneous?

2.      In line 23, the authors say “LF photography can capture multi-view and multi-focal information of a 3D scene”.

LF cameras do not capture multi-focus information. It has to be retrieved computationally.

3.      Lines 37-40, “Despite these advantages, since 3D information of a microscopic sample is obtained using MLA, it is greatly affected by the lighting environment, and since the depth of field and viewing angle are tiny, it is difficult to visualize with satisfactory image quality [8].”.

Why do the authors say that the depth of field is tiny? The depth of field of an LF microscope is much bigger than a conventional microscope.

It is not clear what they mean by "viewing angle" nor what they mean by "visualize with satisfactory image quality", if they are talking about LF microscope. In addition, they reference a paper on a particular LF photographic camera, which makes the explanation more confused.

4.      Lines 48-49, “the only way to improve the resolution is to increase the number of elemental images [16], a quantity that is proportional to the resolution.”.

This is not the only way. In fact, the best way to increase the resolution of an LF microscope is to use the Fourier light field microscope, in which the MLA is placed at the Fourier plane of the microscope objective. In this system, the resolution is increased decreasing the number of elemental lenses that fit into the diameter of the exit pupil.

5.      Lines 69-71, “Inserting the MLA into the image plane of the microscope, as shown in Figure 1, allows all rays from a given location in the sample to be stored as an elemental image array (EIA).”.

Figure 1 does not represent the optical scheme of an LF microscope. In fact, the tube lens is missing.

6.      Lines 79-82, “In IP, the lateral resolution is spatial and the angular resolution can be defined as the number of element lenses (Ns × Nt) and the number of pixels (Nu × Nv) corresponding to each elemental lens.”.

This is not the definition of angular resolution.

7.      Lines 82-83, “The total resolution (Nu × Nv × Ns × Nt ) of LFM depends on the number of resolvable sample spots in the specimen.”.

(Nu × Nv × Ns × Nt ) is not the total resolution, but only the total number of pixels and, obviously, this quantity does not depend on the resolvable sample spots.

8.      Lines 83-85, “In the intermediate plane, the resolution is given by: (Eq. 1) where M is the magnification and λ is the wavelength of illumination.”.

In an LF microscope such as that of Levoy et al. (to which the authors seem to refer), the geometrical term for the resolution is not negligible with respect to the diffraction term. In fact, the geometrical term usually determines the final resolution.

9.      Lines 85-87, “The relationship between the spatial resolution and the angular resolution is (Eq. 2) where W × H is the dimension of the lens array.”.

This is obvioulsy wrong. If Nu, Nv are the number of pixels corresponding to each elemental lens, as stated above, they cannot depend on the resolution as defined in Eq. 1.

10.   Lines 87-88, “The commonly accepted measure of axial resolution is the DOF.”.

This is false. The DOF is the axial range in which the resolution loss is less than square root of 2. The axial resolution is the minimum axial distance between two resolvable points.

11.   Lines 105-108, “The reason for the usage of different LFMs is because of the magnification: the SZX7-based LFM acquires the 3D information of a specimen through low magnification, and the BX41-based LFM is used for high magnification.”.

Both the system used are different systems and have different optics. Olympus SZX7 is a stereomicroscope while Olympus BX41 is a conventional infinity-corrected optical microscope. The authors should clarify how they adapted the system to work with both microscopes.

12.   In Table 1.

From the data of Table 1, I have many doubts about the system.

In an LF microscope such as that of Levoy et al., the f-number (or NA) of the microlenses must be matched to the f-number (or NA) of the objective in the image space, to avoid overlapping between the microimages or the loss of information (microimages too small).

From the data shown, the NAs are not matched, above all in the system with Olympus SZX7.

Moreover, the data for the objective of the low magnification microscope says 10x, but I think the objective used has 1x magnification. In fact, no 10x objective exists for Olympus SZX7.

13.   Line 112, “The EIAs include 76 × 76 elemental images each”.

Why 76x76 elemental images?

Each side of the sensor is 15.52 mm, which divided by the pitch of the microlenses gives 124 images. What is the source of this discrepancy?

14.   Lines 176-177, “Firstly, the point cloud object is obtained from the estimated depth data and high-resolution 2D image,”.

Even if the depth map is merged with the high-resolution 2D image, the resolution of the depth map is very poor (76x76 pixels). I guess the depth map was resized to the dimensions of the 2D image and interpolated. Even so, the 3D point cloud has an actual poor resolution. To obtain a 3D image with good quality in the LF display, it is fundamental to capture the sample with an optimized system. The LF microscope system implemented by the authors is not optimized. The images obtained have a very low quantity of pixels (76x76) as well as a very low optical  resolution (for the high magnification system, it is approximately 25 microns).

15.   In Figure 7.

The authors should have compared the results with those obtained with the method presented in this paper:

Palmieri, Luca, et al. "Robust depth estimation for light field microscopy." Sensors 19.3 (2019): 500.

This paper presents a method for the depth estimation of samples captured with Fourier light field microscope (but it is applicable to other LF systems too) and it obtains very good results.

Moreover, it also exploits the depth estimation for the generation of images for LF displays, through a back-propagation algorithm applied to a point cloud obtained merging the color and depth information of the samples, which is basically the same as the paper proposed by the authors.

16.   In Figure 7(f),

Why is the black area outside the circle of the real image not matted? It creates artefacts in the 3D model and it is an easily resolvable issue.

Author Response

Authors’ responses to Reviewer 1’s comments

Journal:            MDPI Sensors

Article:            sensors-2189347

Title:     High-Quality 3D Visualization System for Light Field Microscopy with Fine-Scale

Shape Measurement Through Accurate 3D Surface Data

Reply to reviewer 1’s comments

The authors sincerely appreciate the reviewer’s detailed insightful and constructive comments and suggestions. The authors have addressed all of your concerns, as described in the point-by-point dialogues below, and mentioned the revisions in the manuscript as underlined texts.

Comment #1:

In the abstract, the authors say “Our approach can simultaneously acquire high-resolution two-dimensional (2D) and light field images”.

But, in Section 3.1, the authors say that the system can acquire the light field images or the 2D images whether the microlens array is inserted in the optical path or not. So, is it really simultaneous?

Reply:

The authors apologize for making the reviewer confused. Currently, due to the completion level of the system, it has a single optical path, so the EIAs and corresponding 2D images were captured while inserting or subtracting the MLA, as mentioned in section 3.1. So we modified some sentences more clearly about it.

However further research considers the system implementation using the only simultaneous capturing method for the EIAs and corresponding 2D images of the specimen as shown in the picture below.

According to the reviewer’s comment, the authors revised the following sentences:

Line # 2–3: “Our approach acquires both high-resolution two-dimensional (2D) and light field images of the specimen sequentially.”

Line # 55–56: “Recently, Kwon et al. used an LFM system to acquire both LF and 2D images.”

  • Line # 303–305: “Both high-resolution 2D images and 4D LF images can be acquired with an existing optical microscope device, and user interaction is applied in visualization by reconstructing a high-quality 3D model.”

Comment #2:

In line 23, the authors say “LF photography can capture multi-view and multi-focal information of a 3D scene”.

LF cameras do not capture multi-focus information. It has to be retrieved computationally.

Reply:

The authors admit to the reviewer’s comment that typical LF cameras cannot capture multi-focus information. However, the LF camera based on integral imaging technique can capture the multi-focus information [Ref. #r1.] Like such LF camera method, the focused-type LFM structure which has been defined in the existing articles (Ref. #5, #6), also can capture the multi-focus information; therefore, the authors used this type of LFM in the proposed system. Figure R1-1 shows the basic structure of the LFM unit of the proposed system. From this analysis, the single object point can be imaged through several elemental lenses, therefore, the proposed system can capture the multi-focus information of the specimen.

Fig. R1-1. Schematic configuration of LFM unit of the proposed system: the sample located at the out-of-focus region is imaged by multiple microlenses[Ref. #r2].

[Ref. #r1: Joo, K. I., Park, M. K., Park, H., Lee, T. H., Kwon, K. C., Lim, Y. T., ... & Kim, H. R. (2019). Light-field camera for fast switching of time-sequential two-dimensional and three-dimensional image capturing at video rate. IEEE Transactions on Industrial Electronics].

[Ref. #r2: Wang, D., Zhu, Z., Xu, Z., & Zhang, D. (2022). Neuroimaging with light field microscopy: a mini review of imaging systems. The European Physical Journal Special Topics].

According to the reviewer’s comment the authors revised the following sentence in lines #23-25:

“LF photography based on the integral photography technique can capture multi-view and multi-focal information of a 3D scene concurrently through a lens array and this can be reconstructed into individual viewpoints or focus images through post-processing.”

Comment #3:

Lines 37-40, “Despite these advantages, since 3D information of a microscopic sample is obtained using MLA, it is greatly affected by the lighting environment, and since the depth of field and viewing angle are tiny, it is difficult to visualize with satisfactory image quality [8].”.

Why do the authors say that the depth of field is tiny? The depth of field of an LF microscope is much bigger than a conventional microscope.

It is not clear what they mean by "viewing angle" nor what they mean by "visualize with satisfactory image quality", if they are talking about LF microscope. In addition, they reference a paper on a particular LF photographic camera, which makes the explanation more confused.

Reply:

The authors agree the reviewer’s comments that the depth of field of an LF microscope is much bigger than a conventional microscope due to each lens of MLA encodes different depth planes. Based on the related researches, the authors also proved this phenomenon a long time ago [Ref. #r3]. However, to the reviewer’s best knowledge, no matter how the depth-of-field of LFM is much wider than conventional optical microscope, it falls lack compared to the actual depth of the sample. That’s why the authors mentioned the depth-of-field of LFM system is “tiny”, in the introduction part.

The term “viewing angle” literally means the field of view of the specimen that is encoded in the EIA. Definitely, more view information of the specimen is obtained by LFM than a general optical microscope, but as the reviewer knows, the viewing angle is also tiny in the case of LFM, because the lack of parallax information of the specimen for the comfortable observation is originally acquired.

Such depth of field and viewing angle are tiny, and the original image quality and resolution are low, so, a fine-quality 3D image cannot be displayed. Therefore, the authors propose that the realistic 3D view reproduction and quality improvement forgery that can satisfy at least general LF cameras and displays, when LFM images are reconstructed as 3D visualizations.

[Ref. #r3: Lim, Y. T., Park, J. H., Kwon, K. C., & Kim, N. (2012). Analysis on enhanced depth of field for integral imaging microscope. Optics Express].

According to the reviewer’s comment, the authors revised the following sentence in line #37-41:

“Despite these advantages, since the 3D information of the specimen is obtained through MLA, the depth of field and viewing angle are encoded tiny, even much better than conventional optical microscopes, and it is greatly affected by the lighting of the environment; therefore, it is difficult to reconstruct the fine-quality 3D visualizations[8].”

Comment #4:

Lines 48-49, “the only way to improve the resolution is to increase the number of elemental images [16], a quantity that is proportional to the resolution.”.

This is not the only way. In fact, the best way to increase the resolution of an LF microscope is to use the Fourier light field microscope, in which the MLA is placed at the Fourier plane of the microscope objective. In this system, the resolution is increased decreasing the number of elemental lenses that fit into the diameter of the exit pupil.

Reply:

The authors appreciate the insightful comment of the reviewer and agree that the Fourier LFM is an efficient way to improve the resolution of LFM system. But in the proposed and related previous systems, the authors used the LFM unit with focused structure. Here, the viewing angle, depth of field, and resolution of the acquired LF image deteriorate as the number of lenses increases; so, if the resolution is improved by reducing the number of lenses, such as Fourier LFM, it can affect to viewing angle and depth of field. Although the authors have increased the depth of field through various methods [Ref. #14,16]; however, the only way to improve the resolution of the LF image obtained by the LFM with such structure has been to increase the number of lenses/number of element images. Therefore, the authors tried to express these meanings in corresponding sentence that in the case of a focused structure LFM, the viewing angle, depth of field, and resolution of the LF image are indicated by the number of lenses, and at this time, to improve the resolution, the number of lenses must be increased.

According to the reviewer’s comment, the authors revised the following sentence in lines #47-50:

“Although various approaches have shown potential for DOF improvement [14,15], the only way to improve the resolution is to increase the number of elemental images [16], a quantity that is proportional to the resolution, when the LFM unit with focused structure is utilized.”

Comment #5:

Lines 69-71, “Inserting the MLA into the image plane of the microscope, as shown in Figure 1, allows all rays from a given location in the sample to be stored as an elemental image array (EIA).”.

Figure 1 does not represent the optical scheme of an LF microscope. In fact, the tube lens is missing.

Reply:

The authors agree with the reviewer’s comment.

According to the reviewer’s comment, the authors revised Fig. 1:

Figure 1. The main acquisition and reconstruction processes of LFM system.”

Comments #6-10:

Lines 79-82, “In IP, the lateral resolution is spatial and the angular resolution can be defined as the number of element lenses (Ns × Nt) and the number of pixels (Nu × Nv) corresponding to each elemental lens.”.

This is not the definition of angular resolution.

Lines 82-83, “The total resolution (Nu × Nv × Ns × Nt ) of LFM depends on the number of resolvable sample spots in the specimen.”.

(Nu × Nv × Ns × Nt ) is not the total resolution, but only the total number of pixels and, obviously, this quantity does not depend on the resolvable sample spots.

Lines 83-85, “In the intermediate plane, the resolution is given by: (Eq. 1) where M is the magnification and λ is the wavelength of illumination.”.

In an LF microscope such as that of Levoy et al. (to which the authors seem to refer), the geometrical term for the resolution is not negligible with respect to the diffraction term. In fact, the geometrical term usually determines the final resolution.

Lines 85-87, “The relationship between the spatial resolution and the angular resolution is (Eq. 2) where W × H is the dimension of the lens array.”.

This is obvioulsy wrong. If Nu, Nv are the number of pixels corresponding to each elemental lens, as stated above, they cannot depend on the resolution as defined in Eq. 1.

Lines 87-88, “The commonly accepted measure of axial resolution is the DOF.”.

This is false. The DOF is the axial range in which the resolution loss is less than square root of 2. The axial resolution is the minimum axial distance between two resolvable points.

Reply:

As the reviewer knows, the author did not analyze the analysis for LFM image processing, because it was already analyzed in detail and precisely by other researchers (M. Levoy et al.[Ref. #6], etc.); and many researchers have recognized it. Similarly, the authors started LFM-based research on basis of this analysis; so, for the authors, it is difficult to judge and take responsibility for whether the points that the reviewer pointed out are true or false. The reason why the authors mentioned this analysis in the paper is to convey the fact that the authors were influenced by these existing analyzes and LFM systems, therefore, the authors titled section 2 as “Principle of Light Field Microscope” and described the basic analysis of LFM imaging, and subsequently, the detailed explanation of the proposed system was expressed.

According to the reviewer’s comment, the authors revised the following sentences in lines #86-88:

“The characteristics of LFM are determined by the lateral and axial resolution [7]. In IP, the lateral resolution is spatial, and the angular resolution can be defined as the number of element lenses and the number of pixels corresponding to each elemental lens.”

The authors removed the following sentences lines #82-88 and Eqs. 1 and 2 from the manuscript:

The total resolution (Nu × Nv × Ns × Nt) of LFM depends on the number of resolvable sample spots in the specimen. In the intermediate plane, the resolution is given by:

                                                                  (1)

where M is the magnification and λ is the wavelength of illumination. The relationship between the spatial resolution and the angular resolution is

                                                                                (2)

where W × H is the dimension of the lens array. The commonly accepted measure of axial resolution is the DOF.

Comment #11:

Lines 105-108, “The reason for the usage of different LFMs is because of the magnification: the SZX7-based LFM acquires the 3D information of a specimen through low magnification, and the BX41-based LFM is used for high magnification.”.

Both the system used are different systems and have different optics. Olympus SZX7 is a stereomicroscope while Olympus BX41 is a conventional infinity-corrected optical microscope. The authors should clarify how they adapted the system to work with both microscopes.

Reply:

In the case of Olympus SZX7, it is suitable for low magnification and has two optical paths; so EIA and 2D images of the sample can be acquired at the same time. However, during the experiment, single optical path was used in both LFM units, because some additional processes such as interconnection and integrated controller, are required to capture the EIAs and 2D images simultaneously. So, the authors planned to complete the additional processes of simultaneous acquisition in the further research. In the case of Olympus BX41, it is suitable for high magnification, but has single optical path. Therefore, it acquires the EIA for the sample when the MLA is inserted; then the 2D image is captured when the MLA is subtracted. The objects grass seed and micro gear are captured by Olympus SZX7-based LFM unit, and the objects balsam seed and fruit fly are captured by Olympus BX41-based LFM unit.

According to the reviewer’s comment, the authors revised the following sentence in lines #106-108:

“The reason for the usage of different LFMs is because of the magnification: the SZX7-based LFM acquires the 3D information of a specimen through low magnification, and the BX41-based LFM is used for high magnification, according to a more suitable condition.”

The authors also revised Fig. 2:

Figure 2. (a) stereo LFM unit for low magnification (top) and biological LFM unit for high magnification (bottom), (b) high-resolution 2D images, (c) corresponding EIAs of the grass seed, micro gear, balsam seed, and fruit fly samples, and (d) a prototype of the 3D LF display unit.”

Comment #12:

In Table 1.

From the data of Table 1, I have many doubts about the system.

In an LF microscope such as that of Levoy et al., the f-number (or NA) of the microlenses must be matched to the f-number (or NA) of the objective in the image space, to avoid overlapping between the microimages or the loss of information (microimages too small).

From the data shown, the NAs are not matched, above all in the system with Olympus SZX7.

Moreover, the data for the objective of the low magnification microscope says 10x, but I think the objective used has 1x magnification. In fact, no 10x objective exists for Olympus SZX7.

Reply:

The authors appreciate the reviewer for discovering and reporting the mistakes in the manuscript. The magnification of the microscope Olympus SZX7 is ×1.

In the case LFM using a photo sensor, the objective NA should be matched to the NA of the MLA.  However, since the proposed system places the DSLR camera, not just the sensor, behind the MLA; so there is no problem obtaining the EIA.

According to the reviewer’s comment, the authors revised Table 1:

“ ”

Comment #13:

Line 112, “The EIAs include 76 × 76 elemental images each”.

Why 76x76 elemental images?

Each side of the sensor is 15.52 mm, which divided by the pitch of the microlenses gives 124 images. What is the source of this discrepancy?

Reply:

Even the MLA consists of 100×100 elemental lenses, 76×76 elemental images have been selected into the region of interest. Because, the outer images are blocked by the MLA plate and did not contain any information (just black images). Therefore, only the images containing the view information of specimen are kept, and outer images are cut out.

According to the reviewer’s comment, the authors revised the following sentence in lines #112-114:

“The EIAs include 76×76 elemental images each after the regions of interest are selected by removing the outer black portions, and the resolution of the entire image is 4000 × 4000 pixels.”

Comment #14:

Lines 176-177, “Firstly, the point cloud object is obtained from the estimated depth data and high-resolution 2D image,”.

Even if the depth map is merged with the high-resolution 2D image, the resolution of the depth map is very poor (76x76 pixels). I guess the depth map was resized to the dimensions of the 2D image and interpolated. Even so, the 3D point cloud has an actual poor resolution. To obtain a 3D image with good quality in the LF display, it is fundamental to capture the sample with an optimized system. The LF microscope system implemented by the authors is not optimized. The images obtained have a very low quantity of pixels (76x76) as well as a very low optical resolution (for the high magnification system, it is approximately 25 microns).

Reply:

The depth estimation method used in the proposed system shows that even in the poor LF images captured through the LFM unit, the boundary, and shape between the background and the object are clearly obtained. And the depth information can be estimated as nearly realistic compared with the previous systems by regenerating a highly reliable sparse depth map to a dense depth map based on matting information. The optical experimental results proved that the proposed system is an appropriate way to estimate the relative realistic depth of field of the specimen when the corresponding LF image is acquired under poor conditions like the LFM units of the proposed system.

So, as the reviewer pointed out, even though the LFM conditions are limited and poor, the authors were able to prove that it is possible for the proposed system to estimate the best depth information and compensate for the lack of textures with 2D images.

Comment #15:

In Figure 7.

The authors should have compared the results with those obtained with the method presented in this paper:

Palmieri, Luca, et al. "Robust depth estimation for light field microscopy." Sensors 19.3 (2019): 500.

This paper presents a method for the depth estimation of samples captured with Fourier light field microscope (but it is applicable to other LF systems too) and it obtains very good results.

Moreover, it also exploits the depth estimation for the generation of images for LF displays, through a back-propagation algorithm applied to a point cloud obtained merging the color and depth information of the samples, which is basically the same as the paper proposed by the authors.

Reply:

The authors appreciate the recommended article, because it is a deeply relevant paper with many points for the authors to refer to. As mentioned in the recommended article, the 3D display is implemented from the estimated depth information; but the proposed method estimates clear shape of the specimen with nearly realistic depth information. In addition, by combining the high-resolution 2D image of corresponding micro object, the proposed system has a feature (or contribution) that it is possible to regenerate a higher quality 3D model; that is slightly different from using the LFM-only images. Depending on the presence or absence of this 2D image information, the quality of the 3D model is determined as shown in Fig. R1-2 below, which also affects the 3D display quality as shown in Figs. 14 and 15 of the manuscript. From the below images, advance in 3D model quality is certified.

Fig. R1-2. The 3D model quality difference between previous (top) and proposed method (bottom).

As the reviewer pointed out, the authors tried compare the sources of both methods; however, meaningful results have not yet been obtained due to compilation problems caused by differences in MATLAB and MEX versions, and in parameters. Also, the authors did not have enough time to make the sufficient comparisons. Future research is underway to use not only focal information, but also correspondence information for LF depth estimation, and solve and reflect the issues which were pointed out. The research the reviewer mentioned is a very meaningful study for us, so we added it as a reference for this paper, and the authors will definitely use it as a comparison study in future studies.

The authors added the following article in the reference section as:

“19. Palmieri, L.; Scrofani, G.; Incardona, N.; Saavedra, G.; Martínez-Corral, M.; Koch, R. Robust depth estimation for light field microscopy. Sensors 2019, 19, 500. ”

Comment #16:

In Figure 7(f),

Why is the black area outside the circle of the real image not matted? It creates artefacts in the 3D model and it is an easily resolvable issue.

Reply:

The proposed method focused on the nearly realistic depth estimation with a clear shape of the specimen and analyzed only the depth estimation results of the specimen area. Likewise, the authors will make sure to reflect the method that the reviewer is suggested, in the further research which is currently in progress.

Thank you once again for reviewing our paper!

Reviewer 2 Report

The Article «High-Quality 3D Visualization System for Light Field Microscopy with Fine-Scale Shape Measurement Through Accurate 3D Surface Datais» about the development of a light field microscopy display system that provides improved image quality.

The authors claim that they put forward a matting Laplacian-based depth estimation algorithm to obtain accurate 3D surface data, allowing the calculation of precise depth data and measurement information from the light field images of specimens (4, 5).

The contradiction lies in the very wording "estimation algorithm" cannot, by definition, give "precise depth data"

The task set by the authors is certainly relevant, but its results must be stated correctly. Meanwhile, the following inaccuracies are visible in the Article:

1.        Intermediate plane (83) is undefined, where is it located?

2.        It is not clear how the authors calculate DOF (88) and how this above calculation is related to the above formulas (1) and (2)? These obvious formulas are not used anywhere at all. Why are they marked? In addition, the parameter A in formula 1 is not even defined

3.       In formulas (4), (5), (6) are not defined (D, d, p, ISet, T).

4.       Next - «which leads to a qualified 3D model without any shortage of information (178)». What is a qualified model? (178) What does it mean «which leads to a qualified 3D model»? (179). Without any shortage for what? Information has been quantified since the last century. It is the humanities who can write about it qualitatively. Representatives of the exact sciences and engineers have long had to speak the language of a quantitative description of a strictly defined quantity - information. How much and why is it needed, how much information is missing? And it never happens that it is not needed at all. Only for the solution of a specific problem it may be enough. And it is necessary to prove it with numbers in hand, if this is so. Even the great Dmitri Mendeleev said: «Science begins as soon as they begin to measure. Exact science is unthinkable without measure».

5.        Again "accurate information" and ensures accurate measurement of the sample (295). How accurate?

6.        And finally, what was seen on the 3D display used by the authors (displayed on the 3D integral imaging display device No. 309). The authors have hidden the most interesting. I think that here it is necessary to place several photos showing the parallax of the restored 3D image (both vertical and horizontal). And if there are no such images, then it is better to completely remove this statement from the text.

In conclusion, I would like to note that the work is interesting, there are results. Only their presentation should be given more attention, focusing on the main goal - understanding the reader, respect for him. I believe that the presented material can be published in the Sensors magazine after completion.

Reviewer 3 Report

This paper presents a light field microscopy display system, composed by the 3d microscope and 3d display.
A matting lapacian helps to get accurate estimates of the depthness of the scene.
Four main examples are presented to validate thair point of view.
One 3d printed test shape was used for calibration proposes.
The results seems interesting and promising, but I have few reamarks to be addressed before gove recommendation for publication on these transactions.
The sentence beggining in the line 59 to 61 must be rewritten because it is confusing.
What is meant with ".. can be expressed in the 5d plenoptic function" in line 73?
It is stated in line 102 that two optical microscopes are used. Which of these two are showed in Figure 2?
Improve Figure 2, including tags and arrows pointing to the things to allow a better compreension. Avoid the placement of text above the photo (for example, the camera is being obstructed by the wrod "ca,era").
Moreove, include a photo showing only the microlens array on at least 2 directions: top and lateral views.
If possible give an estimation of the cost of the complete ML system.
What is the size of matting laplacian matrix that was usesd?
In the 3d board, rather than printing in 3d, did you have conjected to use acrylic and paint to improve the roughness?
Moreover, how reliable and robust is the calibration process for bright and reflexive surfaces?

Reviewer 4 Report

The comments on the manuscript entitled "high-quality 3D visualization system for light field microscopy with fine-scale shape measurement through accurate 3D surface data" by Ki Hoon Kwon et al.:

This manuscript presents a light field microscopy display system that provides improved image quality and accurate three-dimensional measurement information. The topic is fair and the results show good improvement in the light field microscopy. In introduction section more recent works should be described and addressed. Also the quality of some figures is poor, such as Figure 13.

I suggest to accept the manuscript for publication in Sensors after minor revisions.

Round 2

Reviewer 1 Report

In the revised version of the manuscript, the authors corrected the part regarding the light field microscope set-up used for the capture of the samples. They now say that they use a focused light field microscope and represent its optical scheme in Figure 1. The method reported can be considered as an improvement in depth calculation of the samples captured with this system, with respect to previous works by the same authors.

The description of the optical system is still confusing, and in many sentences the authors reference the work of Levoy et al., in which a different light field microscope was proposed, with the microlens array placed at the image plane of the microscope. The working principles of light field microscopy might result unclear to a reader that is not expert in this particular field.

The results of the depth estimation, as said in the first revision, are good, especially considering that the optical system for the capture provides small-size and low-resolution images of the samples. Still, there are some unclear points about the method for the 3D image generation, as no details are given about how the small-sized low-resolution depth image was adapted to the high-size high-resolution 2D image from conventional microscope.

There are still some important issues that have to be addressed before considering the paper publishable, which I report below.

-        In reply to comment 3, the fact that a single object point can be imaged through several microlenses, does not mean that the LF camera/microscope captures multi-focus information. LF systems capture angular information, from which it is possible to calculate depth information of the object.

-        The authors now state that they have used a set-up completely different from the one they depicted in the first version of the submitted paper. They changed totally Figure 1, inserting a tube lens and displacing the MLA far from the image plane. The authors state that they used the system proposed in ref. 6, by Levoy et al., but this set-up does not correspond to the light field microscope proposed in ref. 6 by Levoy et al.

-        Lines 37-41: it is still unclear what the authors mean.

-        Lines 75-77: “Inserting the MLA into the image plane of the microscope, as shown in Figure 1, allows all rays from a given location in the sample to be stored as an elemental image array (EIA).”. As the authors now changed Figure 1 and the set-up used for the capture, the MLA is not inserted at the image plane of the microscope as they say in these lines.

-        The authors do not report the analysis on LF microscopy made in ref. 6, as they say in the reply to comments 6-10. The analysis made in the mentioned reference is correct, while the definitions reported in the first version of the submitted paper are not. Besides, the system used in this work is not the same as that of ref. 6. In lines 86-88, the definitions are still very confusing. Generally speaking, the explanation of the working principles of the light field microscope can result confusing to a reader that is not expert in this field.

-        In lines 112-114, they should clarify that the removed black portions were due to the MLA plate.

-        In reply to comment 14. I agree in that using a high resolution 2D image can result in visually better results, above all if compared to an extremely low resolution image. Still, in a 3D display system, the quality of the reconstructed 3D image depends also on the quality of the depth information, that in the case of the system proposed is quite low. Merging a low resolution depth image with a high resolution 2D image does not create a high definition 3D model, even if the 2D images are visually good.

The information about how the low resolution depth map was adapted to the high resolution 2D image is still missing. Was the depth map rescaled to the size of the 2D image? Or how was it done?

Besides, there are some issues when adapting the image captured through the conventional microscope to the image captured with the light field microscope. One of them is the depth of field. The depth of field of the conventional microscope is much smaller than that of the LF microscope. Does this mean that this method is applicable only to samples whose depth does not exceed the depth of field of the conventional microscope?

-        Information about the computation time of the depth image is missing.

-        In lines 282-283, “it can be verified that the quality was as high as the 2D images”. The quality of the reconstructed 3D image cannot be compared to the captured 2D image.
